Regulation of antimycin biosynthesis by the orphan ECF RNA polymerase sigma factor σAntA

Seipke Ryan F. r.seipke@leeds.ac.uk
Patrick Elaine
Hutchings Matthew I. m.hutchings@uea.ac.uk
School of Biological Sciences, University of East Anglia , Norwich Research Park , Norwich , United Kingdom
Souza Valeria
Electronic publication date: 2014 Feb 6
Publication date: 2014
Volume: 2
Electronic Location ID: e253
Received 2013 Oct 25; Accepted 2014 Jan 7
Copyright: © 2014 Seipke et al.
Copyright year: 2014
Copyright holder: Seipke et al.
License: This is an open access article distributed under the terms of the Creative Commons Attribution License, which permits unrestricted use, distribution, and reproduction in any medium, provided the original author and source are credited.
License URL: https://creativecommons.org/licenses/by/3.0/

Keywords: Streptomyces, Antibiotics, Secondary metabolites, Actinomycetes, Gene regulation, ECF sigma factor, Antimycin

Funding: This work was supported by the Medical Research Council (Milstein award G0801721), the Natural Environment Research Council (grant NE/J01074X/1) and the John and Pamela Salter Charitable Trust. The funders had no role in study design, data collection and analysis, decision to publish, or preparation of the manuscript.

==============================
Antimycins are an extended family of depsipeptides that are made by filamentous actinomycete bacteria and were first isolated more than 60 years ago. Recently, antimycins have attracted renewed interest because of their activities against the anti-apoptotic machineries inside human cells which could make them promising anti-cancer compounds. The biosynthetic pathway for antimycins was recently characterised but very little is known about the organisation and regulation of the antimycin (ant) gene cluster. Here we report that the ant gene cluster in Streptomyces albus is organized into four transcriptional units; the antBA, antCDE, antGF and antHIJKLMNO operons. Unusually for secondary metabolite clusters, the antG and antH promoters are regulated by an extracytoplasmic function (ECF) RNA polymerase sigma factor named σAntA which represents a new sub-family of ECF σ factors that is only found in antimycin producing strains. We show that σAntA controls production of the unusual precursor 3-aminosalicylate which is absolutely required for the production of antimycins. σAntA is highly conserved in antimycin producing strains and the −10 and −35 elements at the σAntA regulated antG and antH promoters are also highly conserved suggesting a common mechanism of regulation. We also demonstrate that altering the C-terminal Ala-Ala residues found in all σAntA proteins to Asp-Asp increases expression of the antFG and antGHIJKLMNO operons and we speculate that this Ala-Ala motif may be a signal for the protease ClpXP.

Introduction

Approximately 60% of the antibiotics and anticancer compounds currently used in human medicine are derived from the secondary metabolites of soil-dwelling Streptomyces species and other filamentous actinomycetes. Although the vast majority of these natural products were discovered more than 40 years ago, the advent of genome mining and new tools to unlock so-called “silent” pathways mean that these bacteria still offer us the best hope of developing new antibiotics for clinical use. The antimycin family of natural products were discovered nearly 65 years ago and initially attracted interest because of their potent antifungal activity (Dunshee et al., 1949). Antimycins are widely produced by Streptomyces species and they exhibit a range of bioactive properties, including antifungal, insecticidal and nematocidal activity. This is the result of their ability to inhibit cytochrome c reductase, an enzyme in the respiratory chain in bacteria and mitochondria. Antimycins are also used as piscicides (brand name Fintrol®) to kill off unwanted scaled fish in the farming of catfish, which are relatively insensitive to antimycins (Finlayson et al., 2002). More recently antimycins have been shown to be potent and selective inhibitors of the mitochondrial Bcl-2/Bcl-xL-related anti-apoptotic proteins which are over-produced by drug resistant cancer cells. Over-production of Bcl-2/Bcl-xL proteins in cancer cells confers resistance to multiple chemotherapeutic agents whose primary mode of action is to trigger apoptosis. Antimycins bind to the hydrophobic groove of Bcl-2-type proteins and inhibit their activity in a mechanism of action that is independent of their activity against electron transport (Tzung et al., 2001). A synthetic derivative of antimycin A3, 2-methoxyantimycin A3 (2-MeAA), no longer inhibits the respiratory chain, but retains potent antagonistic activity toward Bcl-2-related proteins and induces apoptosis (Tzung et al., 2001; Schwartz et al., 2007). This has led to suggestions that antimycin derivatives such as 2-MeAA could be used alongside traditional apoptosis-inducing chemotherapeutics to block drug resistance and kill cancer cells. There is significant interest in bioengineering antimycins with improved pharmacological properties for the treatment of cancer and infectious diseases.

Despite their unique chemical structure and important biological properties, the antimycin biosynthetic pathway was only reported very recently (Seipke et al., 2011a, Seipke et al., 2011b) and rapid progress has been made in elucidating the biosynthetic steps in this pathway over the last two years (for a recent review see Seipke & Hutchings, 2013). Antimycins are produced by a hybrid non-ribosomal peptide synthetase (NRPS)/polyketide synthase (PKS) assembly line for which the complete biosynthetic pathway has been proposed (Sandy et al., 2012; Yan et al., 2012). The AntFGHIJKLN proteins encode the biosynthetic pathway for the unusual starter unit, 3-aminosalicylate-CoA (Schoenian et al., 2012; Sandy et al., 2012). The AntCD proteins comprise the hybrid NRPS/PKS machinery, and AntE and AntM are crotonyl-CoA reductase and discrete ketoreductase homologues, respectively (Sandy et al., 2012). AntO and AntB are tailoring enzymes. AntO is predicted to install the N-formyl group (Yan et al., 2012; Sandy et al., 2012), and AntB is a promiscuous acyltransferase that catalyses a transesterification reaction of a hydroxl group at C-8 to result in the acyloxyl moiety and the chemical diversity observed at R1 (Sandy et al., 2013). The antA gene encodes an extracytoplasmic function (ECF) RNA polymerase sigma (σ) factor named σAntA which, like all other ECF σ factors, contains only two of the four σ70 domains (Staron et al., 2009).

The resurgence of interest in the biosynthesis of antimycins and particularly in engineering new analogues with better pharmacological properties led us to investigate the transcriptional organisation and regulation of the antimycin gene cluster. The only regulator encoded by the ant gene cluster is σAntA, but regulation of secondary metabolite clusters by ECF σ factors is unusual and has not yet been reported in Streptomyces species. To our knowledge only two examples of ECF σ factor regulation of antibiotic biosynthesis have been described and both differ from σAntA because they are co-encoded with, and regulated by, anti-σ factors whereas σAntA is an orphan, i.e., it has no co-encoded anti-σ factor. The two known examples both control lantibiotic production in rare actinomycetes. In Microbospora corallina, the pathway specific regulator MibR and the ECF σMibX regulate microbisporicin biosynthesis and σMibX is regulated by MibW (Foulston & Bibb, 2010). In Planomonospora alba the pathway specific regulator PspR, the ECF sigma factor σPspX and its anti-σ factor PspW all regulate production of the lantibiotic planosporicin (Sherwood & Bibb, 2013). The closest homologues to σMibX and MibW are σPspX and its anti-σ factor PspW, suggesting a common mechanism of regulation for these lantibiotics.

Here we characterize the gene organization of the antimycin gene cluster and the role of σAntA in Streptomyces albus S4. We report that σAntA is regulated at the transcriptional level and controls production of the unusual precursor 3-aminosalicylate that is required for antimycin production. We also show that σAntA represents a new sub-family of ECF σfactors that are only found in the ant gene clusters of Streptomyces species and provide evidence that suggests σAntA regulation of the divergent antGF and antHIJKLMNO operons is conserved in all antimycin producing strains. Finally we provide preliminary evidence that the activity of σAntA is affected by the two C-terminal amino acid residues such that altering the natural Ala-Ala residues to Asp-Asp increases expression of the σAntA target genes. Since a C-terminal Ala-Ala motif is a well known signal for the serine protease ClpXP (Flynn et al., 2003) this may provide a novel post-translational mechanism for controlling σAntA activity without the need for an anti-σ factor.

Materials and Methods

Growth media and strains. Streptomyces strains (Table 1) were grown on mannitol-soya flour (MS) agar and Lennox broth (LB) (Kieser et al., 2000), and Escherichia coli strains (Table 1) were grown on LB or LB agar. Growth media was supplemented with antibiotics as required at the following concentrations: apramycin (50 μg/ml), carbenicillin (100 μg/ml), hygromycin B (50 μg/ml), kanamycin (50 μg/ml), nalidixic acid (50 μg/ml). All Streptomyces strains were created using cross-genera conjugation in which DNA was transferred from E. coli ET12567/pUZ8002 (MacNeil et al., 1992) according to standard methods (Kieser et al., 2000).

Table 1 Strains, cosmids and plasmids used in this study.

Strain		Reference	
S4	Wild type Streptomyces albus S4	Barke et al., 2010	
S4 ΔantA	S4 antA null mutant; AprR	This study	
S4 antA attBΦBT1::pIJ10257-antA	Complemented antA null mutant; antA transcription driven by the ermE* promoter; AprR, HygR	This study	
S4 ΔantA attBΦBT1::pAU3-45-antA-AA	Complemented antA null mutant; antA driven its native promoter; AprR TspR	This study	
S4 ΔantA attBΦBT1::pAU3-45-antA-DD	Complemented antA null mutant with mutations A172D A173D with antA driven its native promoter; AprR TspR	This study	
S4 ΔantC	S4 antC null mutant; AprR	This study	
S4 ΔSTRS4_02194	S4 STRS4_02194 null mutant; AprR	This study	
S4 ΔSTRS4_02195	S4 STRS4_02195 null mutant; AprR	This study	
S4 ΔSTRS4_02212	S4 STRS4_02212 null mutant; AprR	This study	
S4 ΔSTRS4_02213-02217	S4 STRS4_02213-02217 null mutant; AprR	This study	
Escherichia coli			
BL21	Host for heterologous protein expression	Novagene	
BW25113	Host for REDIRECT PCR targeting system	Gust et al., 2003	
DHM1	Host for Cya bacterial two hybrid system	Karimova et al., 1998	
ET12567	Non-methylating host for transfer of DNA into Streptomyces spp. (dam, dcm, hsdM); CamR	MacNeil et al., 1992	
TOP10	General cloning host	Invitrogen	
VCS257	Host strain for Gigapack III XL phage	Agilent Technologies	
Cosmids			
Supercos1	Cosmid backbone for S. albus S4 cosmids; AmpR, KanR	Stratagene	
Cosmid 213	Supercos1 derviative containing the entire antimycin gene cluster; AmpR, KanR	This study	
Cosmid 213 ΔantB-flp	Cosmid 213 derivative containing an 81 bp scar in place of the antB gene	This study	
Cosmid 456	Supercos1 derviative containing a portion of the antimycin gene cluster; AmpR, KanR	This study	
Plasmids			
pCRII-TOPO	Cloning vector for PCR products; AmpR, KanR	Invitrogen	
pAU3-45	pSET152 derivative, integrates into ΦC31 attB site in Streptomyces; AprR TspR	Bignell et al., 2005	
pAU3-45-antA-AA	pAU3-45 derivative containing the antA gene, an 81 bp ‘scar’ in place of antB, and 270 bp upstream of antB cloned into the EcoRI site	This study	
pAU3-45-antA-DD	pAU3-45 derivative containing the antA gene encoding A172D and A173D mutations, an 81 bp ‘scar’ in place of antB, and 270 bp upstream of antB, cloned into the EcoRI site	This study	
pET28a	Protein expression vector; KanR	Novagene	
pET28a-antA	pET28a derivative containing the antA coding sequence cloned into the NdeI and HindIII sites	This study	
pGEM-T Easy	Cloning vector for PCR products; AmpR	Promega	
pIJ773	PCR template for aac3(IV)+ oriT cassette used in REDIRECT PCR targeting system	Gust et al., 2003	
pIJ790	Encodes lambda RED recombination machinery induced by arabinose; CamR	Gust et al., 2003	
pIJ10700	PCR template for hygR cassette used in REDIRECT PCR targeting system	Gust et al., 2003b	
pIJ10257	pMS81 derivative containing ermE*p, integrates into the ΦBT1 attB site in Streptomyces; HygR	Hong et al., 2005	
pIJ10257-antA	pIJ10257 derivative containing the antA coding sequence cloned into the NdeI-HindIII sites	This study	
pUZ8002	Encodes conjugation machinery for mobilization of plasmids from E. coli to Streptomyces; KanR	MacNeil et al., 1992	
Notes.

Amp ampicillin

Apr apramycin

Hyg hygromycin

Kan kanamycin

Cam chloramphenicol

Cosmid library construction and screening. A Supercos1 cosmid library was constructed from Streptomyces albus S4 genomic DNA partially digested with Sau3AI and packaged into Gigapack III XL phage according to the manufacturer’s instructions (Agilent Technologies). One thousand cosmid clones were screened by PCR using primers RFS172 and RFS173 (Table S2), which target an internal fragment of the antC gene. Cosmid 456 and cosmid 213 tested positive and were end-sequenced using primers RFS184 and RFS185 (Table S2) and mapped onto the Streptomyces albus S4 genome using BLAST 2.2.23+ (Altschul et al., 1990).

Construction of Streptomyces albus S4 mutant strains. Mutant strains were constructed using λ-RED based PCR-targeting mutagenesis (Gust et al., 2003). A disruption cassette consisting of a conjugal origin of transfer (oriT) and the apramycin resistance gene, aac(3)IV from pIJ773 (Gust et al., 2003), was generated by PCR using BioTaq polymerase (Bioline) and oligonucleotide primers (Table S2) containing 39 nt of homology that included the start and stop codons of each gene (with the exception of the STRS4_02213-02217 multi-mutant) and 36 nt upstream or downstream of the open reading frame. The resulting PCR products were gel purified and electroporated into E. coli BW25113/pIJ790 harboring either cosmid 456 (ΔSTRS4_02194, ΔSTRS4_02195, ΔantA, ΔantC) or cosmid 213 (ΔSTRS4_02222, ΔSTRS4_02213-STRS4_02217). Transformants were screened for the presence of mutagenised cosmid by NotI digestion. Mutagenised cosmids were moved to S. albus S4 by conjugation. Transconjugants were selected for apramycin resistance and kanamycin sensitivity. The integrity of mutant strains was verified by PCR using flanking primers for each deleted coding sequence together and in combination with the P1 and P2 primers which target the apramycin cassette (Gust et al., 2003). Combinations RRF228 and 229, RRF278 and 279 and RRF329 and 330 were used to test the 02194, 02195 and 02212 knockouts, respectively (Table S2).

Construction of plasmids. In order to heterologously express and purify AntA, the antA coding sequence was PCR-amplified from genomic DNA using oligonucleotide primers engineered to possess NdeI and HindIII restriction sites (RFS230 and RFS 231, Table S2) using Phusion polymerase (New England Biolabs). The resulting PCR product was gel purified and digested with NdeI and HindIII (Roche) and ligated with pET28a (Novagene) cut with the same enzymes using T4 DNA ligase (Promega) to create pET28a-antAI. DNA sequencing using the T7 promoter and T7 terminator primers (Novagene) verified the integrity of the cloned antA coding sequence. In order to construct the antA over-expression/complementation plasmid, pIJ10257-antA, the antA coding sequence was excised from pET28a-antA using NdeI and HindIII and ligated to pIJ10257 (Hong et al., 2005) cut with the same enzymes. pIJ10257-antA was introduced into Streptomyces strains by conjugation and transconjugants were selected for resistance for hygromycin.

In order to generate complementation constructs in which transcription of wild-type and mutated antA was initiated by its native promoter, we replaced the antB gene with an apramycin resistance cassette using the REDIRECT system described above using oligos RFS188 and RFS189 (Gust et al., 2003, Table S2). The apramycin cassette possesses two FRT sites recognised by the FLP recombinase. The mutagenised cosmid was introduced into E. coli strain BT340, which expresses a FLP recombinase when cultured at 42°C (Gust et al., 2003). FLP recombinase-mediated excision of the apramycin resistance cassette leaves an 81 bp in-frame “scar.” Cosmid 213 ΔantB-flp was used as template for PCR with the forward primer RFS351 and the reverse primers RFS231 or RFS352 (Table S2). RFS351 targets 270 bp upstream of the putative antB start codon, and RFS231 and RFS352 both target an identical sequence in the C-terminus of antA, with the exception that RFS352 introduces two C → A point mutations, which introduces A172D and A173D changes into the resulting AntA protein. These PCR products were cloned into pGEMT-Easy (Promega) and verified by DNA sequencing using M13R and M13F oligonucleotides. Next, the antA-containing inserts were excised from pGEMT-Easy by EcoRI digestion and ligated with pAU3-45 (Bignell et al., 2005) digested with the same enzyme. pAU3-45-antA-AA and pAU3-45-antA-DD were introduced into Streptomyces strains by conjugation and transconjugants were selected for resistance to thiostrepton.

Phylogenetic analysis. Antimycin gene clusters were analysed from S. ambofaciens ATCC 23877 (AM238663, (Choulet et al., 2006)), S. blastmyceticus NBRC 12747 (AB727666, (Yan et al., 2012)), S. gancidicus BKS 13-15 [AOHP00000000, (Kumar et al., 2013), S. griseoflavus Tü4000 (ACFA00000000), S. hygroscopicus subsp. jinggangensis 5008 (NC_017765), S. hygroscopicus subsp. jinggangensis TL01 (NC_020895), Streptomyces sp. 303MFCol5.2 (ARTR00000000), Streptomyces sp. TOR3209 (AGNH00000000, (Hu et al., 2012) S. albus S4 (CADY00000000, (Seipke et al., 2011b)), S. albus J1074 (NC_020990), Streptomyces sp. SM8 (AMPN00000000), Streptomyces sp. NRRL2288 (JX131329), (Yan et al., 2012)), Streptomyces sp. LaPpAH-202 (ARDM00000000), Streptomyces sp. CNY228 (ARIN01000033). σAntA proteins were aligned to five (when possible) random proteins from each ECF RNA polymerase σ factor subfamily defined by Staron et al (Staron et al., 2009) by using Clustal (Sievers et al., 2011). The phylogenetic tree was created using PhyML 3.0 with the default settings (Guindon et al., 2010) and visualised using FigTree v1.4 (http://tree.bio.ed.ac.uk/software/figtree/).

HPLC analysis. Wild-type and mutant strains were cultured atop a cellophane disc on MS agar. At the time of harvest, the cellophane disc containing mycelia was removed and either processed for RNA extraction (below) or discarded. Bacterial metabolites were extracted from the spent agar using 50 ml of ethyl acetate for 1 h. 20 ml of ethyl acetate was evaporated to dryness under reduced pressure and the resulting residue was resuspended in 400 μl 100% methanol. In all cases, the methanolic extracts from at least two biological replicates were mixed and centrifuged at >16,000 g in a microcentrifuge prior to analysis. Antimycin A1–A4 standards were purchased from Sigma-Aldrich. 35 μl of methanolic extract was separated on a Phenomenex C18(2) 5 μm 4.6 × 150 mm using a Hitachi L-6200 HPLC system and the following gradient (solvent A: water, solvent B: methanol, flow rate 1 ml/min): 0–20 min, 10–100% B; 20–34 min 100% B; 34.1–44 min, 10% B. Samples were analysed with a Shimadzu M20A Photo Diode Array.

RNA analysis. For all experiments involving RNA, S. albus S4 strains were cultivated at 30°C on MS agar atop a cellophane disc to facilitate the easy harvest of mycelia into microcentrifuge tubes. Transcription was arrested using a stop solution (95% ethanol, 5% acid phenol) diluted 1:4 with water. Total RNA was extracted using a RNeasy Mini Kit (Qiagen) according to the manufacturer’s instructions and included both an on-column and a post-column DNaseI treatment. The absence of DNA contamination was assessed by PCR. DNase-treated RNA was reverse transcribed using 250 μg of random hexamers and Superscript III reverse transcriptase (RT, Invitrogen) with an extension temperature of 55°C.

For co-transcription analysis, twenty-nine cycles of PCR amplification with six primer sets (Table S2) were performed using cDNA originating from 5 μg of RNA with BioTaq Polymerase (Bioline). Primer sets were designed to span the intergenic regions of the antimycin cluster and targeted at least 300 bp upstream of putative start codons to account for promoters driving transcription from multiple sites within a transcriptional unit. RNA from the complemented antA mutant strain (ΔantA/pIJ10257-antA) was used, because transcript abundance was greater for operons involved in 3-aminosalicylate biosynthesis. The PCR products obtained were cloned into either pCRII-TOPO (Invitrogen) or pGEM-T Easy (Promega) and sequenced by either the Genome Analysis Centre (Norwich, UK), Source BioScience (Cambridge, UK), or Eurofins MWG Operon (Ebersberg, Germany) using oligonucleotide primer M13r (Integrated DNA Technologies).

For quantitative RT-PCR, gene-specific primers were designed to amplify ∼100 bp from the first and last gene of each transcriptional unit in the antimycin cluster. cDNA was diluted (1 volume of cDNA to 2 volumes of water) and target genes were quantified using a Bio-Rad CFX96TM instrument and SensiFastTM SYBR No-ROX kit (Bioline). Each treatment consisted of three biological replicates and two technical replicates. The calculated Ct (threshold cycle value) for each target gene was normalized to the Ct obtained for the hrdB gene, which encodes the vegetative sigma factor and is routinely used as a reference gene for transcriptional analyses (Kelemen et al., 1996).

For mapping of transcriptional start sites, 10 μg of RNA from the ΔantA/pIJ10257-antA strain was processed using the FirstChoice® RLM-RACE Kit (Ambion) according to the manufacturer’s instructions with the following modifications: for cDNA synthesis, Superscript III RT (Invitrogen) was used to according the manufacturer’s instructions using an extension temperature of 55°C. The gene-specific primers used for each transcriptional unit are listed in Table S2. The final PCR products were gel purified and cloned into pCRII-TOPO (Invitrogen) or pGEM-T Easy (Promega) and sequenced using oligonucleotide primers M13r (Integrated DNA Technologies) by either the Genome Analysis Centre (Norwich, UK), Source BioScience (Cambridge, UK) or Eurofins MWG Operon (Ebersberg, Germany). The transcriptional start site was determined to be the nucleotide immediately adjacent to the sequence of the 5′RLM-RACE RNA adapter.

Bacterial two-hybrid analysis. The full STRS4_02195, AntA, and STRS4_04339 (SigB orthologue) coding sequences were PCR-amplified from S4 genomic DNA using Phusion Polymerase (New England Biolabs) and primers RFS280 and RFS281 (STRS4_02195), RFS282 and RFS283 (antA), and sigB (RFS284 and RFS285) (Table S1). The gel purified PCR products were digested with BamHI and KpnI (Roche) and cloned into bacterial two-hybrid plasmids pUT18C and pKT25 (Karimova et al., 1998) cut with the same enzymes. Cloned inserts were sequenced by The Genome Analysis Centre (Norwich, UK) using primers RFS286, RFS287 (pUT18C clones) and RFS288 and RFS289 (pKT25 clones) to ensure that no mutations had occurred. Plasmid combinations of interest were co-electroporated into E. coli DHM1 and processed for β-galactosidase activity as previously described (Hutchings et al., 2002).

Results and Discussion

Organisation and expression of the antimycin gene cluster

To facilitate mutagenesis of the antimycin gene cluster, we constructed a Supercos1 library of the S. albus S4 genome (Genbank accession CADY00000000.1) and screened the library by PCR against an internal fragment of antC. We identified two overlapping cosmids containing antC and confirmed that cosmid 213 contains the complete predicted ant gene cluster by deleting genes adjacent to the cluster using PCR-targeted mutagenesis (Fig. 1). To define the upstream border we deleted STRS4_02194, which encodes a separate NRPS and STRS4_02195 which encodes a predicted membrane protein of unknown function. To determine the downstream border we deleted STRS4_02212 and STRS4_02214-STRS4_02217 which are predicted to encode a nitrate/nitrite assimilation protein and an ABC-transport system, respectively. To determine if these mutations affect antimycin production we performed bioassays against the human pathogen Candida albicans and observed no obvious difference in the ability of the S. albus S4 strains to inhibit the growth of C. albicans compared to wild-type (Fig. 2A). High performance liquid chromatography (HPLC) confirmed that antimycin production is not affected by any of these mutations showing that STRS4_02194, STRS4_ 02195, STRS4_02212, and STRS4_02214-02217 mark the boundaries of the ant gene cluster (Fig. 2B). The gene organization of the ant cluster suggests there is a minimum of four transcriptional units with the largest being the antHIJKLMNO operon (Fig. 1). Almost all of these ORFs overlap, suggesting transcriptional and translational coupling, but as a proof of principle we confirmed that the antGF and antHIJKLMNO genes are co-transcribed by performing end-point RT-PCR. Six primer pairs were designed to span the intergenic (or overlapping gene) regions of the antGF and antHIJKLMNO operons and targeted at least 300 bp upstream of the putative start codons to detect transcriptional read-through. Six PCR products were obtained by RT-PCR analysis and sequenced to confirm that antGF and antHIJKLMNO form two operons. No products were obtained when reverse transcriptase was omitted (Fig. S1). In addition to confirming that antGF and antHIJKLMNO are organized into operons, this also validates our approach to analysing their expression using qRT-PCR to measure mRNA levels of the first and last genes in each operon.

Figure 1 The antimycin biosynthetic gene cluster in Streptomyces albus S4.

Genes shaded grey indicate those that are required for antimycin biosynthesis. Genes shaded black were experimentally determined not to be required for antimycin biosynthesis. Narrow black arrows indicate the presence of four operons and the direction of their transcription. The locations of cosmid 213 and cosmid 456 are indicated by horizontal lines and the double vertical hash indicates that cosmid 456 is comprised of additional DNA that falls outside the boundaries of this schematic.

Figure 2 Defining the boundaries of the antimycin gene cluster.

(A) Streptomyces albus S4 WT and mutant strains challenged with Candida albicans. Null mutants of genes adjacent to the gene cluster (coloured black in Fig. 1) produce an antimycin-positive phenotype, characterised by a large circular zone of cleared C. albicans growth. The ΔantC mutant strain displays an antimycin-negative phenotype, but retains residual antagonistic activity against C. albicans due to the production of candicidin, a second antifungal compound produced by this strain (Barke et al., 2010; Seipke et al., 2011a). (B) High-performance liquid chromatography (HPLC) of metabolites produced by S. albus S4 WT and mutant strains. The ΔantC mutant does not produce antimycins, while null mutations in genes adjacent to the antimycin cluster had no effect on antimycin production.

Streptomyces species have a complex life cycle that includes growth as a substrate mycelium that gives rise to aerial mycelia and sporulation. To determine at which stage of the life cycle the antimycin gene cluster is expressed we measured expression of the four ant operons after 18 and 42 h growth on mannitol-soya flour (MS) agar. After 18 h growth on MS agar S. albus S4 consists entirely of substrate mycelium but after 42 h the substrate mycelium has differentiated to produce aerial mycelium and spores. All four ant operons are expressed at a significantly higher level at 18 h (in substrate mycelium) compared to 42 h which suggests that all four ant operons are switched off following differentiation (Fig. 3A). Conversely, HPLC analysis of mycelium and culture medium extracted at the same time points only detected antimycins in the 42 h samples suggesting there is a lag between ant gene expression and antimycin production (Fig. 3B). This is probably due to the time it takes for the precursor to be produced and for the antimycin scaffold to be assembled and then accumulate to detectable levels. Most notably, these data suggest that specific regulatory mechanisms exist to activate ant gene expression in substrate mycelium and switch it off again following differentiation. Since antA is the only putative regulatory gene in the ant gene cluster we investigated the role of σAntA in regulating antimycin production.

Figure 3 There is a delay between expression of the antimycin biosynthetic genes and the production of antimycins.

(A) HPLC analysis of metabolites produced by S. albus S4 wild-type. Antimycins are detected in media extracts of 42 h old but not 18 h old cultures. (B) qRT-PCR analysis of the antimycin gene cluster in 18 and 42 h old cultures shows that expression of the antimycin gene cluster is significantly down-regulated following differentiation. *** denote that values reported are statistically significantly different with a P value < 0.001 in a Student’s T-test.

Antimycin production is dependent on the orphan ECF sigma factor σAntA

To investigate the role of σAntA in regulating antimycin biosynthesis, we deleted the antA gene and tested the mutant strain against C. albicans in a bioassay. The antA mutant is significantly less active against C. albicans compared to wild-type and this is consistent with loss of antimycin production (Seipke et al., 2011a). Complementation of this mutant with the antA gene under the control of the strong constitutive ermE* promoter restores bioactivity against C. albicans to wild-type levels (Fig. 4A) and HPLC analysis confirmed that antimycins are not produced by the antA mutant (Fig. 4B). We conclude that σAntA is required for antimycin production.

Figure 4 σAntA is required for the biosynthesis of antimycins.

(A) S. albus S4 strains challenged with Candida albicans. The ΔantA null mutant shows dramatically reduced bioactivity compared to the wild-type strain and the complemented strain (ΔantA/pIJ10257-antA). The residual bioactivity of the ΔantA mutant is due to the continued production of candicidin, a second antifungal compound. (B) HPLC analysis of metabolites produced by S. albus S4 strains. Antimycins were only detected in extracts prepared from the wild-type and the ΔantA/pIJ10257-antA strains, and not the ΔantA null mutant.

To determine which of the four ant promoters are regulated by σAntA we used qRT-PCR to measure ant operon expression in the wild-type and antA strains grown for 18 h on MS agar. Deletion of antA did not affect the level of transcription of either the antBA or antCDE operons, but transcription of both the antGF and antHIJKLMNO operons was significantly reduced in the antA mutant (Fig. 5). This suggests that σAntA positively regulates the transcription of the antFGHIJKLMNO genes which encode biosynthesis of 3-aminosalicylate, the precursor used by the AntC NRPS. Furthermore, over-expression of σAntA in 42 h cultures activates the expression of the antGF and antHIJKLMNO operons leading us to conclude that no additional regulators are required to activate the antG and antH promoters (Fig. 6). In addition the antB and antC promoters must be regulated by a transcription factor encoded outside of the ant gene cluster since they are upregulated at 18 h relative to 42 h growth. To confirm this we introduced cosmid 213 into S. lividans, S. coelicolor M145 and the S. coelicolor superhost strains M1146, M1152 and M1154 (Gomez-Escribano & Bibb, 2011) but failed to detect antimycin production, supporting the idea that at least one additional transcription activator is required.

Figure 5 σAntA activates transcription of the antFG and antHIJKLMNO operons.

qRT-PCR analysis of antimycin genes in the wild-type and ΔantA strains after 18 h growth. Transcription of antFG and antHIJKLMNO is significantly reduced in the ΔantA mutant strain, whereas transcription of antBCDE are unaffected. *** denote that values reported are statistically significantly different in a Student’s t test with a P value < 0.001 in a Student’s T-test.

Figure 6 σAntA alone is sufficient to activate transcription of antFG and antHIJKLMNO in 42 h old cultures.

qRT-PCR analysis of wild-type or ΔantA/pIJ10257-antA in 42 h old cultures shows that repression of σAntA-regulated genes can be overcome by over-expressing antA.

σAntA and its putative binding site are highly conserved

Bioinformatic analysis failed to identify the common ECF σ factor promoter motifs upstream of the antG and antH genes, notably the “AAC” motif in the −35 region and the “CGT” motif in the −10 region (Staron et al., 2009). We therefore mapped the transcriptional start sites of the antGF and antHIJKLMNO operons using 5′-RLM RACE and identified −10 and −35 regions which share high nucleotide sequence identity with one another, but not with the σAntA-independent antB promoter (Fig. 7A). Six antimycin producing Streptomyces strains have been reported previously (Riclea et al., 2012; Seipke et al., 2011a; Yan et al., 2012) and we identified eight more putative antimycin gene clusters whilst searching for σAntA orthologues in Genbank (Table S1 and Experimental Procedures). Since the 14 known σAntA orthologues share 66% sequence identity (Table S1, Fig. S2), we hypothesise that σAntA regulation of the antG and antH promoters will be common to all antimycin producing Streptomyces strains. To investigate this, we searched for the antG and antH promoter motifs in the 14 known or predicted antimycin gene clusters encoded by published Streptomyces genome sequences. All 14 antG promoters contain very high sequence identity in the −35 and −10 regions, although S. ambofaciens has an 18 nucleotide spacer between the −35 and −10 element compared to the typical 17 nucleotide spacer (Fig. 7B). High nucleotide conservation was also observed at the antH promoter and the −10 element contains a “CTC” motif that is 100% conserved across all promoters although again spacer regions between the −10 and −35 elements vary in length between 17 and 18 bp (Fig. 7B). The in silico data therefore suggests that σAntA has highly conserved −35 and −10 binding sites at the antG and antH promoters of all antimycin producing Streptomyces strains. Scanning the complete published S. albus genome with the AntA −10 and −35 binding sites (using GLEME2 – part of MEME) returns only two significant hits, the antGF and antHIJKLMNO promoters suggesting there are no other σAntA targets (results not shown) (Bailey et al., 2009). This strongly suggests that σAntA is a pathway-specific regulator of antimycin biosynthesis.

Figure 7 Identification of σAntA promoter motifs.

(A) The −10 and −35 motifs at the σAntA-target promoters of antFG and antHIJKLMNO are nearly 100% identical and display zero nucleotide identity with the promoter region of antAB. Shared identity is indicated by grey shading. The nucleotides mapped by 5′RLM-RACE is denoted by +1 and are shown in bold face (B) Conservation of the S. albus S4 antFG and antHIJKLMNO promoter elements in other antimycin-producing Streptomyces species. Conservation between the experimentally determined promoter region of S. albus S4 and the putative promoter regions of other antimycin producers is indicated by grey shading. (C) Consensus sequence for the −35 and −10 promoter elements recognised by σAntA displayed as a WebLogo (Crooks et al., 2004). Below are the full strain names and accession numbers for antimycin-producing strains: S. ambofaciens ATCC 23877 (AM238663), S. blastmyceticus NBRC 12747 (AB727666), S. gancidicus BKS 13-15 (AOHP00000000), S. griseoflavus T4000 (ACFA00000000), S. hygroscopicus subsp. jinggangensis 5008 (NC_017765), S. hygroscopicus subsp. jinggangensis TL01 (NC_020895), Streptomyces sp. 303MFCol5.2 (ARTR00000000), Streptomyces sp. TOR3209 (AGNH00000000), S. albus S4 (CADY00000000), S. albus J1074 (NC_020990), Streptomyces sp. SM8 (AMPN00000000), Streptomyces sp. NRRL2288 (JX131329), Streptomyces sp. LaPpAH-202 (ARDM00000000), Streptomyces sp. CNY228 (ARIN00000000).

Figure 8 Altering the terminal Ala-Ala motif of σAntA results in higher expression of σAntA targets.

The ΔantA null mutant was complemented with either a wild-type copy of antA or a variant of antA encoding A172D and A173D point mutations. After 42 h of growth, transcription of σAntA-targets in the ΔantA/antA-DD strain were significantly greater than both the wild-type and ΔantA/antA-AA strains, suggesting the terminal Ala-Ala motif modulates stability and/or activity of σAntA. ** and *** denote that values reported are statistically significantly different with a P value < 0.01 and 0.001 in a Student’s T-test, respectively.

σAntA represents a new sub-family of ECF sigma factors

σAntA contains only the σ2 and σ4 domains (Pfam families PF04542 and PF08281) which is characteristic of the ECF family of RNA polymerase σ factors (Staron et al., 2009). However, σAntA does not fit into any of the ECF sub-families listed in the well-maintained public database ECF Finder (Staron et al., 2009). Multiple sequence alignments of the 14 σAntA homologues in the database and representatives of all known ECF sub-families revealed that the σAntA proteins form a distinct clade and therefore represent a new sub-family of ECF σ factors (Table S1 and Fig. S3). ECF σ factors are rare in secondary metabolite gene clusters and to our knowledge this is the first example in Streptomyces species (Foulston & Bibb, 2010; Sherwood & Bibb, 2013). The only obvious candidate for an anti-σAntA factor in the antimycin gene cluster is the putative membrane protein STRS4_02195. However, it is absent from the ant clusters in other streptomycetes, its removal has no effect on antimycin biosynthesis (Fig. 2) and it does not interact with σAntA in bacterial two-hybrid analysis (Fig. S4) which leads us to conclude that σAntA is an orphan ECF that is not subject to anti-σ factor control. However, since antA expression is activated in substrate mycelium (by an as yet unknown regulator) and switched off following differentiation we predict that a mechanism might exist to remove σAntA protein at this stage of growth. The only unusual feature in the primary sequence of the 14 σAntA homologues is the conserved C-terminal Ala-Ala (AA) motif (Fig. S2) which is a known signal for the serine protease ClpXP (Flynn et al., 2003). To test whether the C-terminal AA residues are required for σAntA activity we made two identical constructs expressed under the control of the native antB promoter. The first construct drives production of the wild-type protein (designated σAntA-AA) and the second drives production of an altered protein in which the AA has been replaced with DD (designated σAntA-DD). We introduced these constructs into the antA mutant and measured expression of the antGF and antHIJKLMNO operons in these strains. Both operons are significantly more highly expressed in the strain producing σAntA-DD compared with the wild type σAntA-AA protein (Fig. 8). These data suggest that the two C-terminal residues play an important role in the stability and/or activity of σAntA and may target σAntA for proteolysis by ClpXP. Unfortunately, all attempts to detect the σAntA protein by immunoblotting whole cell extracts with polyclonal anti-σAntA antibodies have been unsuccessful while tagging the protein at the N-terminus inactivates the protein (not shown). Future work will therefore be required to determine the role of the C-terminal AA motif in σAntA however, we have provided preliminary evidence that suggests σAntA might be a direct target for ClpXP, thereby bypassing the requirement for the additional level of anti-σ regulation. This would also provide a rapid mechanism to shut down precursor biosynthesis when antimycins are no longer required.

Supplemental Information

Supplemental Information 1 Supplementary Information

This document contains two supplementary tables and four supplementary figures as well as legends for each of these.

Click here for additional data file.

We thank Charles Brearley for assistance with HPLC, Barrie Wilkinson, Mervyn Bibb and Mark Buttner for helpful comments on this work and all members of the Hutchings group and the UEA iGem 2013 team for useful discussions.

Additional Information and Declarations

Competing Interests

Author Contributions

The authors declare that they have no competing interests.

Ryan F Seipke conceived and designed the experiments, performed the experiments, analyzed the data, contributed reagents/materials/analysis tools, wrote the paper.

Elaine Patrick performed the experiments, contributed reagents/materials/analysis tools.

Matthew I Hutchings conceived and designed the experiments, analyzed the data, contributed reagents/materials/analysis tools, wrote the paper.

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
