# Peer review of "Regulation of antimycin biosynthesis by the orphan ECF RNA polymerase sigma factor σAntA"

_PeerJ, doi:10.7717/peerj.253_

## Round 0.1 · original submission · Minor Revisions

This is a very nice work on molecular biology and fine regulation of the antimycin biosynthesis gene cluster from Streptomyces albus. The findings are very interesting and the text just need a final polishing.

Reviewer 1 ·

Basic reporting

No comments

Experimental design

No comments

Validity of the findings

This manuscript describes the organization of the antimycin biosynthesis gene cluster from Streptomyces albus. The authors show that all the genes in the cluster are organized as four transcriptional units and that the only regulatory gene localized inside the cluster is that for an ECF sigma factor, sigma-antA. The authors show that sigma-antA belongs to a new subfamily of ECF sigma factors, all of which are localized inside gene clusters for antimycin biosynthesis from other streptomycetes. Two of the transcriptional units inside the antimycin cluster are dependent on sigma-antA for transcription, and conserved sequences, presumed to be important for binding by this novel ECF sigma factor to the promoter, are identified. The authors also suggest that sigma-antA activity might be regulated by the ClpXP protease, and provide some preliminary indirect evidence for this speculation.
This is an interesting manuscript that presents the first example of a Streptomyces secondary metabolite gene cluster regulated by an ECF sigma factor. The data presented are of sufficient quality for publication. However there are some minor points that should be addressed by the authors before the manuscript can be accepted.
1. The authors identify the borders of the antimycin cluster by obtaining mutant strains with deletions in neighboring genes, both upstream and downstream the ant cluster. The upstream border was identified by deleting genes STRS4_02194 and STRS4_02195, and the downstream border by deleting genes STRS4_02212 and STRS4_02214-STRS4_02217. None of these deletions has an effect on antimycin production. No mention is made of STRS4_02213. Is there a reason why this gene was left intact in these experiments? Why was it not considered for deletion instead of genes further downstream? In this respect Figure 1 should be improved, as there are only four black arrows to the right of STRS_02212 but these are labeled as corresponding to five genes (02213-02217). One of these black arrows is misaligned.
2. The authors should state in the figure legends what the error bars in figures 3, 5, 6, 8 and S4 represent (I assume it is the standard deviation).
3. It is incorrect to describe operons antAB and antFG as such, since the gene order should correspond to the direction of transcription. These two operons should be referred to as the antBA and antGF operons throughout the text and in Figure S1 (B)
4. Figure 8 describes the effect of replacing the terminal Ala-Ala motif at the C-terminus of sigma-AntA on expression of the target operons. The authors should discuss the statistically significant increase in antH and antO expression levels (relative those of the wild type strain, i.e. gray and white bars) when the (antA-AA) construct is used to complement the antA deletion strain, particularly since this is not observed for the antF and antG transcripts. This difference is much higher in the case of the antO transcript levels, yet the figure marks this difference as statistically less significant (two asterisks, P<0.01) than that for antH (three asterisks, P<0.001). Are the authors sure about these analyses?
5. Figure S2, the blue shading should be corrected to include the DVL amino acids in the sigma4 domain of the S. ambofaciens AntA orthologue.
6. The paragraph from lines 116 to 132 should be rewritten to improve clarity (i.e. it described the generation of two constructs not just “a construct”). This should be explained more clearly.

Reviewer 2 ·

Basic reporting

This is a well written manuscript that describes the regulation of antimycin biosynthesis in Streptomyces albus by a novel transcriptional activator, named by the authors as sigma-AntA. It should be of significant interest to those interested in bacterial gene regulation and in natural product biosynthesis.

Experimental design

The experimental approaches and methods are entirely appropriate, and the work carried out to a high standard.

Validity of the findings

The results and overall conclusions are certainly worthy of publication in PeerJ, but I have a few minor criticisms/questions that should be addressed/answered before acceptance for publication (particularly point 2):

1. Summary, line 3: Presumably this sentence should read: “activities against the ANTI-apoptotic machineries inside human cells”, not “the apoptotic machineries”.

2. Lines 47 and 287-293: The authors claim that AntA belongs to the ECF family of sigma factors, but the only data that substantiate this are presented in the SI, which is a pity. Apparently, the AntA only shows similarity to ECF sigma factors in conserved regions 2 and 4. Could the authors state the level of amino acid identity/similarity in these regions to experimentally verified ECF sigma factors, or show an alignment to substantiate this point? Currently, it is not clear to me why the authors believe that AntA belongs to the ECF family of sigma factors. A phylogenetic tree is shown in Figure S3, but the text is too small to be able to see where the Ant family of putative ECFs sits with respect to known sigma factors. And how “significant” is this clustering? I believe that this statement needs further justification. However, in my view, the paper would still be worthy of publication even if a convincing bioinformatic case could not be made that AntA belongs to the ECF family.

3. Line 228: The use of a RT-PCR primer corresponding to sequences “300 bp upstream of the putative start codon” does not eliminate the possibility of a transcriptional start site located in the region of interest (“line 228 – “to exclude possible intragenic promoters”). RT-PCR will only indicate transcriptional read-through. The text needs to be modified accordingly.

4. Line 248: At this stage of the work, this ought to read “Since antA is the only PUTATIVE regulatory gene in the ant gene cluster…..”. i.e. “Putative” should be inserted.

5. Line 264: “artificially” does not seem to be a very appropriate term. “precociously” or “prematurely” better?

Additional comments

No additional comments.

Reviewer 3 ·

Basic reporting

No comments

Experimental design

Lines 102-103: please be more specific about what primer combinations were used to verify gene knockouts. Not all PCRs are equally diagnostic…

Validity of the findings

The conclusion that no regulators outside of sigma-AntA are needed for expression of the antFG and antH-O operons seems reasonable – however, one cannot exclude the possibility that sigma-AntA has other target genes outside of the antimycin cluster, and the products of these may contribute to cluster activation. There are two relatively straight-forward experiments that would help to address this possibility. The first is searching the S. albus genome for other sigma-AntA-like promoters (given the very strong consensus sequence generated by the authors). There isn’t any obvious resistance gene encoded in the cluster (and given that antimycins inhibit bacterial cytochrome c reductase, presumably some sort of resistance mechanism is required? – a pump maybe?) – could this be under the control of sigma-AntA elsewhere in the chromosome?

The second experiment would involve taking advantage of cosmid 213 which appears to contain the entire antimycin biosynthetic cluster, and moving this into a heterologous host like S. coelicolor M1152. Successful expression of antimycin would support the idea that all necessary genes are contained on this cosmid (although it would not exclude the possibility that global transcription regulators may contribute to antimycin expression). If this was attempted but failed to result in antimycin production, this might suggest that other regulatory (or metabolic) elements are required – but would be useful information to include regardless. To this reviewer, this is an important experiment to include in a revised manuscript.

Additional comments

In the submitted manuscript, Seipke and colleagues describe the first in-depth genetic characterisation of the antimycin biosynthetic cluster. They effectively mapped the ends of the biosynthetic cluster (although previous bioinformatics analyses had provided strong support for the experimentally determined boundaries), and determined that cluster activity required the AntA sigma factor.

1. Antimycin expression: transcript levels for the different gene clusters were assessed after 18 (vegetative) or 42 (aerial/spores) h. However in S. coelicolor, secondary metabolism typically (or at least historically is thought to) initiates during the onset of aerial hyphae formation. Investigating a timepoint between 18 and 42h (when aerial hyphae formation is initiating) may show expression that is higher still – or it may not, which in itself would be interesting (as it would shake up this dogma a bit).

2. It would be helpful to include information about where the transcription start points are for the antG and antH clusters relative to their translation start sites, and how large the intergenic region is separating these two operons.

3. Line 295: replace “potential” with “obvious”. What is it about STRS4_02195 that makes it a reasonable anti-sigma factor candidate? There exist orphaned anti-anti-sigma factors (e.g. BldG), so it may still be possible that sigma-AntA has a cognate anti-sigma factor, but that it is found elsewhere in the chromosome. Furthermore, on the ‘StrepDB’ website, there is another (small) gene annotated in the region between antB and antC (02198). Is this gene transcribed along with antA and antB? (indicating the exact transcription start site for this operon would be useful as well) Could this possibly encode an anti-sigma factor?

4. Figure 3: there is an enormous difference in expression levels between antF and antG. Any speculation as to what is going on there?


Editorial comments:

Line 25: what hydrophobic groove is being discussed? Why is antimycin binding here significant?

Lines 48-49 and 53-54: repetitive

Line 74: please change “ClpX” to “ClpXP”, and provide an accompanying reference

Throughout: micro, delta, gamma and degree symbols failed to convert properly in the PDF

Line 82: italicise “E. coli”

Line 109: instead of “T4 DNA polymerase” did the authors mean “T4 DNA ligase”?

Line 96: please remove the rogue bracket

Line 270: mention that transcriptional start sites were mapped using 5’-RLM RACE

Line 276: does “sequence identity” refer to amino acid sequence identity?

Table 1: the spacing of the plasmid information makes it challenging to figure out what information corresponds to any given plasmid

For all RT-PCR results, what do the error bars represent? Standard deviation? Standard error?

Figure 1: change to “cosmid 456 comprised additional DNA”

Figure 7: the “zero nucleotide identity” shared between antAB and the AntA-targeted promoters is modestly overstated, as there are two nucleotides that are shared between these. Perhaps replace “zero” with “little”. The first two sentences in section (B) are also completely repetitive.

---

## Round 0.2 · accepted · Accept

the manuscript has all the suggested changes, it is now fit for publication